# An Improved MobileNet Network with Wavelet Energy and Global Average Pooling for Rotating Machinery Fault Diagnosis

**DOI:** 10.3390/s22124427

**Published:** 2022-06-11

**Authors:** Fu Zhu, Chang Liu, Jianwei Yang, Sen Wang

**Affiliations:** 1Key Laboratory of Advanced Equipment Intelligent Manufacturing Technology of Yunnan Province, Kunming University of Science & Technology, Kunming 650500, China; 20202203144@stu.kust.edu.cn (F.Z.); yjw326101@163.com (J.Y.); wangsen0401@kust.edu.cn (S.W.); 2Faculty of Mechanical & Electrical Engineering, Kunming University of Science & Technology, Kunming 650500, China

**Keywords:** fault diagnosis, lightweight network, pooling method, wavelet convolution

## Abstract

In recent years, neural networks have shown good performance in terms of accuracy and efficiency. However, along with the continuous improvement in diagnostic accuracy, the number of parameters in the network is increasing and the models can often only be run in servers with high computing power. Embedded devices are widely used in on-site monitoring and fault diagnosis. However, due to the limitation of hardware resources, it is difficult to effectively deploy complex models trained by deep learning, which limits the application of deep learning methods in engineering practice. To address this problem, this article carries out research on network lightweight and performance optimization based on the MobileNet network. The network structure is modified to make it directly suitable for one-dimensional signal processing. The wavelet convolution is introduced into the convolution structure to enhance the feature extraction ability and robustness of the model. The excessive number of network parameters is a challenge for the deployment of networks and also for the running performance problems. This article analyzes the influence of the full connection layer size on the total network. A network parameter reduction method is proposed based on GAP to reduce the network parameters. Experiments on gears and bearings show that the proposed method can achieve more than 97% classification accuracy under the strong noise interference of −6 dB, showing good anti-noise performance. In terms of performance, the network proposed in this article has only one-tenth of the number of parameters and one-third of the running time of standard networks. The method proposed in this article provides a good reference for the deployment of deep learning intelligent diagnosis methods in embedded node systems.

## 1. Introduction

In mechanical equipment, key components often fail due to high speed and heavy load, variable working conditions and other harsh working conditions, resulting in the failure of the entire transmission system and even major safety accidents. Real-time condition monitoring [1] and fault diagnosis of key components of machinery equipment are important to ensure the safe and smooth operation of mechanical systems.

Traditional fault diagnosis techniques are usually signal processing methods based on manual feature extraction, such as Fourier transform [2], wavelet transform [3], empirical mode decomposition [4] and other methods to extract features to achieve fault diagnosis. Zhang et al. [5] used the kurtosis index and correlation coefficient to construct measurement indexes. They use the maximum weighted kurtosis index to analyze sensitive patterns and finally achieve fault diagnosis by extracting fault features. Han et al. [6] combined the Teager energy operator and the signal processing method of complementary ensemble empirical mode decomposition (CEEMD) to effectively extract the fault features of bearings and identify bearing faults. The above methods require artificial extraction of features, which are greatly affected by human factors and the diagnosis is highly dependent on expert experience and prior knowledge. With the development of computer and network technology, various machine learning [7] algorithms represented by supervised learning and unsupervised learning have been successfully applied in the field of fault diagnosis. Compared with the later research on deep learning methods, the machine learning methods at this time are usually also called shallow learning methods. Pandya et al. [8] used the Hilbert–Huang transform and the asymmetrical proximity function APF-KNN method to mine acoustic emission signal information. They used this method to achieve high-precision classification and diagnosis of bearings. Cao [9] used the PCA dimensionality reduction method combined with the support vector machine theory to establish a three-layer fault classification model. He realized the diagnosis quickly and accurately in the nuclear field. Although the above shallow machine learning methods can achieve good diagnostic results, the results still depend on the quality of manual feature extraction, and they do not really realize automatic diagnosis.

The deep learning [10] method has been widely used in the field of fault diagnosis by virtue of its strong feature learning and extraction ability between layers. It has successfully solved the fundamental problems of traditional diagnosis technology and shallow machine learning models, such as large human interference, low efficiency and weak learning ability, showing good performance. Long et al. [11] used a modified autoencoder (SAE) to enhance the one-dimensional vibration signal. They converted the augmented data set into 2D images and put them into a CNN for training, realizing fault diagnosis in the case of many types of faults and a small number of data sets. Xu [12] used continuous wavelet transform to transform vibration signals into time-frequency images. In addition, he used CNN to extract features from images and input the extracted features into the gcForest classifier for diagnosis. It achieves higher accuracy and robustness than the single network model. Zhong [13] proposed a transfer learning [14] method for gas turbine fault diagnosis based on CNN and SVM, which demonstrated the model’s good learning and transferability under the condition of small samples. Hasan et al. [15] proposed an interpretable artificial intelligence fault diagnosis model in the field of bearing fault diagnosis. The model has a good interpretable architecture and good generalization ability and can perform bearing fault diagnosis under variable working conditions. However, the above methods can effectively realize intelligent diagnosis of key components of mechanical equipment through the combination of different networks and algorithms. They overcome the difficulties and low efficiency of traditional signal processing methods and shallow machine learning feature extraction. However, with the improvement of diagnostic accuracy, the neural network becomes larger and more complex. The hardware resources required for model training and prediction are gradually increasing, so neural network models tend to run only on servers with high computing power. This makes it difficult for trained deep learning models to be deployed in light-resource embedded systems or mobile terminals, which challenges the application of deep learning methods in the current engineering field.

In order to effectively reduce the size of the model and improve the efficiency of model inference, the lightweight and automation of deep learning networks is a current research hotspot. Since 2016, lightweight network architectures, such as SqueezeNet [16], ShuffleNet [17], NasNet [18] and MobileNet have been proposed successively, whereas MobileNet is representative of a lightweight neural network. The study found that, compared with AlexNet [19], VGG16 [20] and other classical networks, MobileNet can greatly reduce the size of the model while ensuring accuracy. MobileNet is currently achieving good results in the field of image recognition. Wang et al. [21] proposed a lightweight lane detection algorithm based on MobileNet + UNet. The algorithm only needs 40 ms for each detection on the PC terminal. Furthermore, it can efficiently and accurately extract lane line features in complex environments to meet the requirements of industrial sites. However, compared with other popular fields, there is less research on MobileNet in the field of fault diagnosis. YU [22] et al. applied MobileNet-V1 to bearing fault diagnosis to realize intelligent diagnosis of bearings. Kim [23] et al. applied MobileNet-v2 to bearing acoustic emission signals to achieve fault diagnosis of bearings based on an embedded system. However, the above two methods need to convert 1D signals to 2D signals and do not allow direct processing of 1D vibration signals. Yao [24] et al. proposed a lightweight bearing fault detection method for butterfly-transform (BFT)-MobileNet V3, which has the advantages of high efficiency and high accuracy and can visualize the quality of extracted features.

Aiming at the problems of common convolution model with many parameters, long operation time, difficulty in running on embedded platform and poor anti-noise performance of the model, an improved MobileNet network lightweight fault diagnosis method based on wavelet energy and GAP (One Dimensional-Wavelet LPPool-GAP-MobileNet, 1D-WL-G-MN) was proposed; the main contributions are as follows:The method is able to process one-dimensional vibration signals directly and does not require other methods to convert one-dimensional signals into two-dimensional signals. Based on the end-to-end idea, the deep learning method is applied to the fault diagnosis of equipment.The introduction of wavelet convolution layers and energy pooling layers enhances the adaptive extraction capability and robustness of the model to features, demonstrating good noise immunity.The impact of using a fully connected layer for fusion classification on the whole network is analyzed, and the use of GAP instead of a fully connected layer for the classification task effectively reduces the number of parameters in the model while ensuring accuracy.The model is deployed in the Raspberry Pi lightweight embedded system, and the applicability of the method on the light-resource embedded platform is verified, which provides a feasible solution for the real-time status online monitoring of key equipment in the development of light-resource embedded systems.

The rest of this article is organized as follows. In the second section, some important background theoretical knowledge is introduced. In the third section, the proposed method, model structure and parameters are described in detail. In the fourth section, the proposed method is validated using the bearing data set of Western Reserve University and the gearbox data set of the QPZZ-II fault simulation test bench. Furthermore, the lightweight deployment of the model in the Raspberry Pi is realized. In Section V, the advantages and disadvantages of the method proposed in this article are summarized, and prospects for future research are proposed.

## 2. Theoretical Background

### 2.1. Wavelet Convolution

The wavelet is a special kind of waveform with a finite length and an average value of zero. At present, the time-frequency method based on Wavelet Transform (WT) has a wide range of applications in rotating machinery fault diagnosis. By constructing a suitable basis function, the wavelet can have the ability to characterize the local characteristics of the signal in the time-frequency domain and detect the transient or singular point of the signal. The essence of wavelet transform is to select wavelet base functions of different scales to perform convolution on the signal, as shown in Formula (1).
(1)S(a,τ)=∫f(t)∗ψ(τ−t,a)dt

Convolutional neural networks are widely used in Natural Language Processing [25] (NLP), one-dimensional time-series signal analysis, signal processing and other fields. It mainly extracts features from layer to layer for learning through the convolution operation between the convolution kernel and the input. The output calculation of the neurons in the convolution layer is shown in Formula (2).
(2)$$Y_{j}^{l}=f(\sum_{i\in M_{j}}(X_{i}^{l-1}*w_{ij}^{l})+b_{j}^{l})$$
where: $Y_{j}^{l}$ is the output of the jth neuron in the l layer and $X_{i}^{l-1}$ is the output of the jth neuron in the l−1 layer; that is, the input of the lth layer, $w_{ij}^{l}$, is the corresponding convolution kernel, $b_{j}^{l}$ is the bias vector of the lth layer and f(x) is the activation function of the l convolution layer. In signal processing, the convolution operation in the time domain is equivalent to the product in the frequency domain, so the convolution of the original signal with the convolution kernel in the time domain is to extract the frequency domain features. When the convolution kernel is regarded as a wavelet basis function of different scales, its convolution with the original signal can achieve the effect of wavelet analysis. Daubechies wavelet is used to construct a series of wavelet convolution kernels to obtain different fault features. Compared with general features extracted from random convolution kernels, wavelet convolution kernels can effectively improve the accuracy of network diagnosis. It has good anti-noise capability and robustness at the same time.

### 2.2. Energy Pooling

In convolutional neural networks, a pooling layer is usually added between adjacent convolutional layers. It can effectively reduce the size of the parameter matrix, thereby reducing the number of parameters in the final task classification layer. At present, the commonly used pooling methods include max pooling, average pooling, energy pooling (LPPool), etc. As an important indicator to judge the operating state of equipment, energy has the characteristics of good stability and repeatability. In order to effectively extract the different frequency band features output by the wavelet convolution layer, the energy pooling method is selected to be used in conjunction with the wavelet convolution. For the signal x, the signal is calculated by the energy pooling method as shown in Formula (3).
(3)$$f(X)=\sqrt[{p}]{\frac{\sum_{x\in X}x^{p}}{N}}$$

In the above formula, when p is infinite, it is equivalent to max pooling; when *p* = 1, it is equivalent to average pooling; Figure 1 shows the calculation results of energy pooling, max pooling and average pooling when *p* = 2, *f* = 2 and *s* = 2, where f is the filter size and s is the stride.

### 2.3. MobileNet Network

The MobileNet [26] network is a lightweight network architecture proposed by Google in 2017. Its biggest feature is the use of depthwise separable convolution to replace the standard convolution in traditional networks. The MobileNet network based on depthwise separable convolution can achieve the same feature extraction function as standard convolution with fewer model parameters. It can reduce network constraints on hardware resources.

The depthwise separable convolution [27] (DS) is composed of depthwise convolution [28] (DW) and pointwise convolution (PW). The structure is shown in Figure 2. DW convolution has only one-dimensional convolution kernels, and each convolution kernel is only responsible for the convolution of one channel. The channels cannot be expanded after the depthwise convolution is completed, and because each convolution operation is performed independently between each channel, the feature information of different channels at the same spatial position cannot be effectively utilized. Therefore, pointwise convolution is required to combine the feature maps generated by the depthwise convolution to generate new feature maps. PW convolution is very similar to standard convolution, except that the size of the convolution kernel is 1 × 1. It will weight the feature map obtained by DW convolution in the depth direction to obtain a new feature map with the same size. It realizes the operation of changing the data dimension with fewer parameters.

### 2.4. Global Average Pooling (GAP)

In the classification task, the traditional convolutional network structure is always followed by one or *N* full connection layers [29] and finally classified in softmax. It is characterized by using a full connection layer to directly flatten the feature map into a one-dimensional vector. It will retain all spatial location information, resulting in an excessive number of model parameters, overfitting and low generalization performance. Assuming that the feature map is an *A* × *A* image with n channels, the full connection layer has *N* neurons, and the parameter calculation of the full connection layer is shown in Formula (4).
(4)P=n×A×A×N+N

Aiming to address this problem, Lin [30] et al. proposed a method of replacing the last fully connected layer with global average pooling (GAP). They generate a feature map corresponding to each category of the classification task; then, they average each feature map, and the resulting vector is directly input to the softmax layer. This process does not have any parameters that need to be optimized, which not only avoids the risk of overfitting brought by the full connection but also realizes the same conversion function as the full connection. In addition, compared with the fully connected black box processing, GAP makes the connection between each category and feature map more intuitive. By summarizing spatial information, the spatial conversion of input is more robust. The effect of full connection and global average pooling is shown in Figure 3.

## 3. The Proposed Fault Diagnosis Method

Faced with the engineering requirements of deploying deep learning models in embedded systems, research on network lightweighting is carried out based on MobileNet. Combining the wavelet convolution energy with the GAP method greatly reduces the complexity of the model and the number of parameters on the premise of ensuring the accuracy. The wavelet convolution kernel is incorporated into the MobileNet network to enhance the mining capability of equipment fault information; then, the energy pooling method is used to extract the band energy information of different channels, and finally, GAP is used to ensure the diagnostic accuracy while reducing the network parameters. The process flow of the proposed method is shown in Figure 4. The data set is preprocessed and divided into a training set and a test set. The training set is fed into the network to train and save the resulting model, and the test set is used to test the performance of the model and finally realize the lightweight deployment of the model in Raspberry Pi.

### 3.1. 1D-WL-G-MN Model Structure

The vibration signal is a one-dimensional signal. It cannot be directly applied to the MobileNet network, so this article builds the MobileNet network based on one-dimensional convolution and using depthwise separable convolution as a module. The network structure is shown in Figure 5. The model is composed of some feature extraction layers and some fusion classification layers. The first layer is a wavelet convolution layer constructed using Daubechies wavelets, followed by an energy pooling layer to extract the frequency band energy information of the different output channels of the wavelet convolution layer. The first two depth-separable convolution layers are followed by a maximum pooling layer to effectively reduce the size of the parameter matrix, and the last depth-separable convolution layer is followed by a global average pooling layer to fuse the extracted features. Finally, the softmax function is used to classify the features to realize the failure classification of key components of mechanical equipment. The specific parameters of the model are shown in Table 1.

### 3.2. Data Preprocessing

In order to expand the training samples under the condition of limited field data, while maintaining the periodicity and continuity of the one-dimensional time series vibration signal to avoid problems such as signal loss caused by equidistant sampling, the data enhancement method of moving sliding window overlapping sampling is used to increase the number of training samples, as shown in Figure 6.

If the total length of data in a certain state is *L*, the data length of each sample is *l*. Using the mobile window overlap sampling enhancement method, the data are sampled with the offset *α*, and the data length of the overlapped part is *l* − *α*. The number of samples that can be divided into the current signal is shown in Formula (5), and the enlarged multiple is shown in Formula (6). In order to make the data in each sample contain sufficient device health information, the offset *α* should be greater than the number of data points collected by the device rotating one circle.
(5)$$B=[\frac{L-l}{\alpha }-1]$$
(6)$$\varphi =\frac{l(L-l+\alpha )}{\alpha L}$$

### 3.3. Model Training

After the data are preprocessed, the data set is divided into the training set and test set according to the ratio of 7:3. The loss function selects the cross-entropy loss function, the optimizer selects the Adam optimizer with better adaptability and more stability, the learning rate is set to 0.001, the decay rate is 0.99, and mini-batch learning is used in the training process (its size is set to 70). The details are shown in Table 2.

## 4. Experimental Analysis and Discussion

### 4.1. Experimental Environment

This article uses the gearbox data set of the QPZZ-II fault simulation test bench and the bearing data set of Western Reserve University to verify the proposed method. The experiments are completed under the following configurations.

Hardware environment: CPU is Intel core i5-10400@2.90 GHz; running memory is 16 G; and GPU is GTX2060.

Software environment: the operating system is Windows 10 64 bit. The programming language is Python3.7. The deep learning framework is Pytorch and the main packages used in this environment include pytorch1.7, numpy 1.19, torchvision 0.8 and h5py 3.1.

### 4.2. Gearbox Experiment of QPZZ-II Fault Simulation Test Bench

The experiment was carried out using the QPZZ-II fault simulation test bench. The structure of the test bench is shown in Figure 7. The device is driven by a motor and consists of a motor, a pulley and a gear box.

The research is carried out with the gearbox as the experimental object. The gear box selects a spur gear meshing pair; the modules of the driving wheel and the driven wheel are both 2, the number of teeth of the driving wheel is 55, and the number of teeth of the driven wheel is 75. We simulated six common fault states of gearboxes on the active spur gears, which are normal, gear broken teeth, local wear, pitting corrosion, tooth profile error and tooth root broken teeth, as shown in the following Figure 8a–e.

The measurement points are positioned at the locations of measurement point 1 and measurement point 2 in Figure 7 of the experimental setup. The signal measured at measurement point 1 is the vibration signal of the active wheel with different fault conditions, and the signal measured at measurement point 2 is the vibration signal of the driven wheel in its normal condition. The ICP accelerometer was used to collect the vibration signal of the gearbox; its model is 603C01, the sensitivity is 100 mv/g and the NI-9234 acquisition card was used to collect the acceleration signal. In the experiment, the motor speed is 600 rpm, the sampling frequency is 25.6 kHz and the number of sampling points is 6000. The time waveform of each fault type vibration signal at measuring point 1 is shown in Figure 9.

The data set is constructed by using the above six gearbox fault states. Data augmentation with moving sliding window overlap sampling was used to increase the training sample size, where the data length for each sample was *l* for 2048 and *α* for 436. There are 1000 samples per class, so there are 6000 samples in total. After data enhancement preprocessing, the data set was divided into a training set and a test set in a 7:3 ratio, with the training set containing 700 × 6 = 4200 samples and the test set containing 300 × 6 = 1800 samples, as shown in Table 3.

In order to intuitively demonstrate the ability of the model to learn features, we used the T-SEN algorithm proposed by Laurens, which reduces the features learned by the global average pooling layer to two dimensions for visual analysis. The visualization results are shown in Figure 10. It can be seen that the model is able to completely separate the various types of signals at large spacing, showing a good classification effect. In addition, we use a confusion matrix to evaluate the performance of the model on the test set. The results are shown in Figure 11. The model can identify 100% of the various samples in the test set, showing good generalization performance.

In order to compare and verify the effectiveness of the proposed method, this article selects multiple networks for comparison, and the network structure is shown in Table 4. The 1D-CNN, 1D-WL-F-MN, 1D-C-F-MN and 1D-C-G-MN are consistent with the methods presented here, except for the differences described in Table 4. For comparison with shallow learning methods, the SVM algorithm is chosen for this article, which uses nine different features for classification: rms, skewness, cliffness, peak, waveform factor, pulse factor, peak factor, margin and mean square value.

The experimental results are shown in Table 5. It can be seen from the experimental results that the fault diagnosis method based on deep learning has great advantages in diagnosis accuracy compared with the SVM machine learning method, and the classification accuracy rate reaches more than 99%. Because the SVM method is quite different from other network training methods in the classification mechanism, the SVM method is not included in the comparative study of the parameter quantity and prediction time of the model. Compared with other comparison methods, the method proposed in this article performs better in terms of accuracy. In terms of model parameters, the number of parameters of the 1D-WL-G-MN model is about 1⁄6 of the fully connected layer fusion classification network 1D-WL-F-MN, which is 1⁄10 of the standard neural network 1D-CNN. In terms of performance, the method proposed in this article is also better than other methods in the time required for prediction, which is 0.4 s less than 1D-WL-F-MN and more than doubled compared to 1D-CNN. Under the premise of ensuring the accuracy, the network is greatly simplified, which fully proves the effectiveness of the method.

The anti-noise performance is an important indicator to judge the quality of a model. In order to verify the anti-noise performance of the proposed method, Gaussian white noise with different signal-to-noise ratios (SNR) was added to the collected original vibration data and compared with the above deep learning methods. The experimental results are shown in Figure 12. The experimental results show that the shallow SVM model is not suitable for signal fault diagnosis under noise interference. The accuracy of the method proposed in this article can always keep above 97% under -6 dB strong noise interference, which is slightly higher than that of 1D-WL-F-MN using a full connection for classification. Compared with the networks that do not use wavelet energy processing, 1D-C-G-MN, 1D-C-F-MN and 1D-CNN have obvious improvements. It is proved that the network improved by introducing wavelet energy has better anti-noise performance.

### 4.3. Western Reserve University Bearing Experiment

To further illustrate the effectiveness of the proposed method, the bearing data set published by Western Reserve University is used for verification. The structure of the test bench is shown in Figure 13.

In this article, the bearing data at the driving end is selected under the condition that the rotational speed is 1750 r/min, the load is 2 HP and the sampling frequency is 12 kHz. The data set contains four types of states: normal, rolling element fault, inner ring fault and outer ring fault, and each fault state corresponds to data of three different fault diameters of 0.007, 0.014 and 0.021 mm, plus the data of the normal state, which is a total of 10 sets of data. Due to the sufficient data of Western Reserve University, this article does not use data augmentation methods to expand the data set. The length of each sample is 2048 points, and the data of each state contains a total of 200 samples, with a total of 200 × 10 = 2000 data samples. The obtained data set is divided into a training set and test set according to the ratio of 7:3, in which the training set has 2000 × 0.7 = 1400 samples, and the test set contains 2000 × 0.3 = 600 samples. The obtained data set is shown in Table 6.

In order to verify the performance of the model on the test set, a confusion matrix was used to analyze the results. The T-SNE algorithm is used to display the features learned by the final global average pooling layer. The results are shown in Figure 14 and Figure 15. It can be seen from the figures that the model can accurately identify each type of data and completely separate various features with large distances.

In this experiment, we compare and analyze 1D-WL-F-MN, 1D-C-G-MN, 1D-C-F-MN, 1D-CNN and SVM. The parameters used are exactly the same as in the previous experiment, except that the six categories of the above experiment are replaced by ten categories. The experimental results are shown in Table 7. For the bearing data set of Western Reserve University, the shallow machine learning SVM algorithm also has a good recognition effect, with an accuracy rate of 89%. The accuracy of deep learning-based methods has reached more than 99%. In terms of performance, with the increase in fault categories, the number of parameters for fusion classification using the full connection layer will increase sharply, whereas the number of parameters using GAP classification is not obvious. The parameter quantity of the method proposed in this article is 1⁄13 of that of the fully connected layer fusion classification network, which is about 1⁄20 of the standard CNN network, and the operating efficiency is improved by nearly three times.

### 4.4. End-to-End Fault Diagnosis Application

As the preferred system of edge node, embedded system has the characteristics of strong portability and low price and has high application value in engineering practice. In order to verify the advantages and field application prospects of the lightweight model proposed in this article, the Raspberry Pi 4B embedded system is selected as the test platform. The CPU of the Raspberry Pi 4B is a 64-bit quad-core 1.5 GHz Broadcom BCM2711, and the memory is 1 GB. The Pytorch environment required to run the model is built on this platform, the compiler is python3.7, the main toolkits of the running environment are Pytorch and Torchvision and their versions are torch1.3 and torchvision 0.4, respectively. We deploy the model trained on the computer to Raspberry Pi, and then test the operating efficiency of different types of data on the deployed model, mainly using the model’s inference time for sample data as a comparison basis. Because processes such as model load time and data import time have the same operation for different models, they are not recorded in the time of comparison. The time for three predictions of the sample was recorded and the average value was calculated. The results are shown in Figure 16. The experimental results show that the model takes about 120 ms for each prediction. The results are better in terms of operational efficiency, compared to fusion classification models using full connectivity (1D-ML-F-MN) as well as standard convolutional neural networks (1D-CNN). Moreover, the experimental results contribute to the fact that it is appropriate to implement intelligent fault diagnosis of critical components in resource-light embedded systems using the proposed method.

## 5. Conclusions, Limitations and Future Research

In addressing the problems of complex neural network structure, large number of parameters and difficulty in deploying in lightweight embedded devices, this article combines wavelet convolutional energy with GAP method and proposes a lightweight MobileNet-based network model for fault diagnosis of mechanical equipment. Drawing on the ability of the wavelet analysis method to mine the time-frequency information of the signal, the wavelet convolution kernel is integrated into the MobileNet network to enhance the mining of equipment fault information. Then, the energy features are obtained from the channel information extracted by the wavelet convolution using the energy pool, which further condenses the ability to represent fault information. The network model is lightweight based on MobileNet, which is convenient for the deployment of engineering applications. The GAP method is introduced to further reduce the network parameters and ensure the diagnosis accuracy. Finally, the Raspberry Pi 4B platform is used to deploy the models trained in this article to verify the accuracy and effectiveness of the lightweight models in fault diagnosis. There are also some shortcomings in this article, as the proposed method does not compare with the original 2D MobileNet network in terms of model complexity, operational efficiency and performance. In addition, there is still room for further research on model performance and efficiency, and further research is needed on model interpretability.

## Figures and Tables

**Figure 1 sensors-22-04427-f001:**
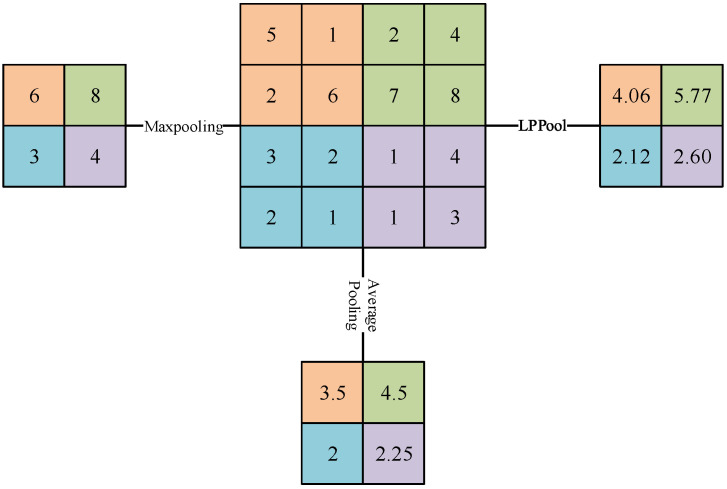
Calculation results of different pooling methods.

**Figure 2 sensors-22-04427-f002:**
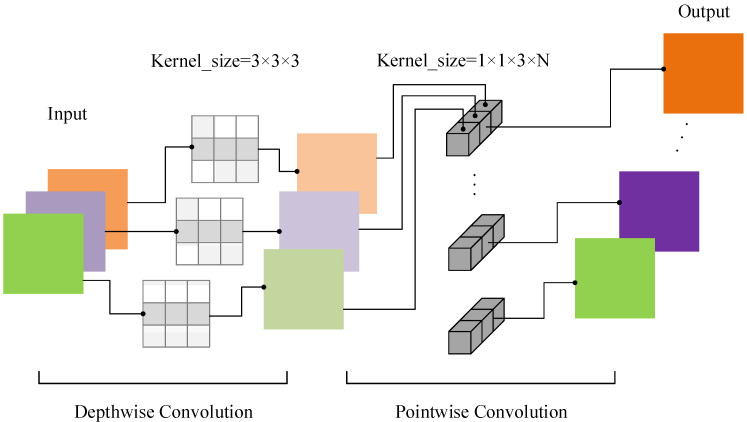
The depthwise separable convolution structure.

**Figure 3 sensors-22-04427-f003:**
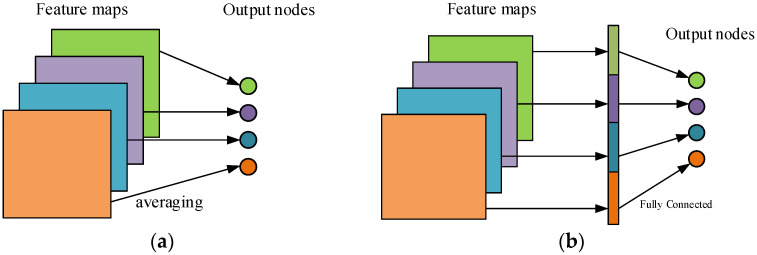
Global pooling versus full connection. (**a**) Global Average Pooling; (**b**) Fully connected Layers.

**Figure 4 sensors-22-04427-f004:**
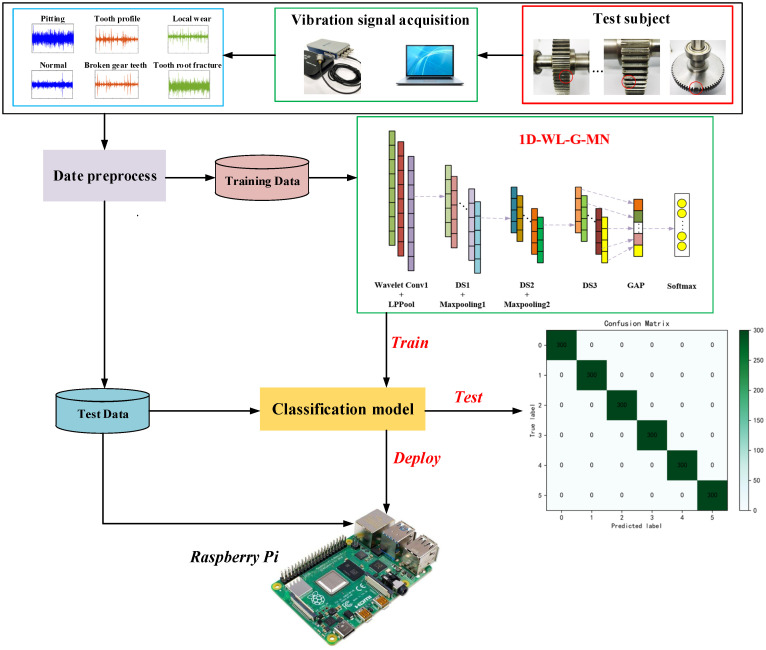
Flow chart.

**Figure 5 sensors-22-04427-f005:**
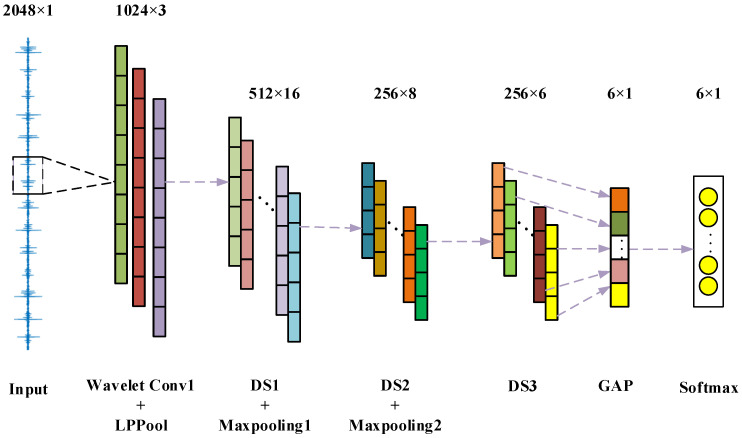
1D-WL-G-MN network structure.

**Figure 6 sensors-22-04427-f006:**
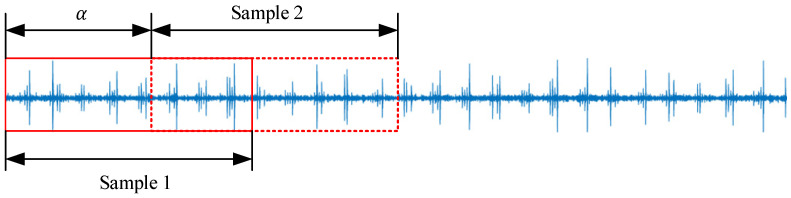
Data enhancement diagram.

**Figure 7 sensors-22-04427-f007:**
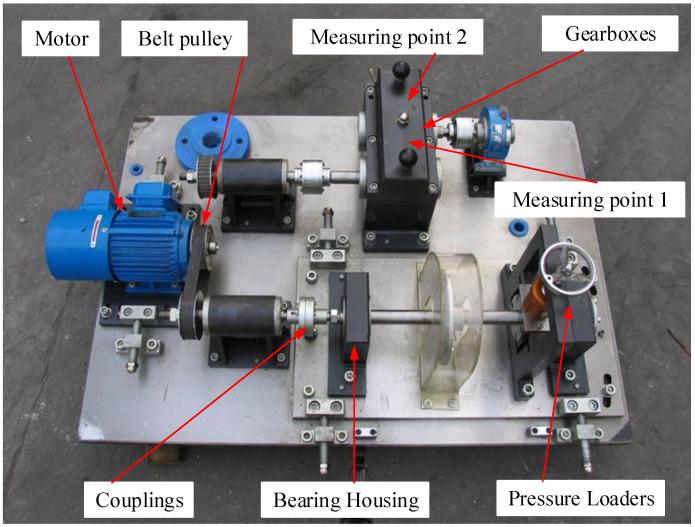
QPZZ-II fault simulation test bench.

**Figure 8 sensors-22-04427-f008:**
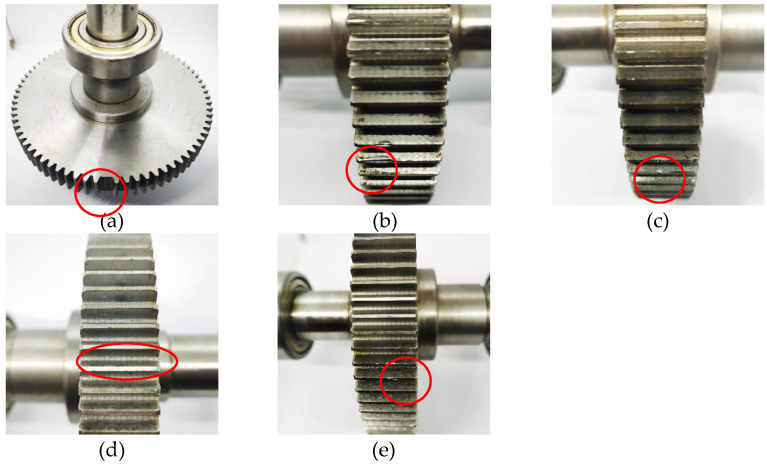
Gear fault diagram. (**a**) Broken gear teeth (**b**) Local wear (**c**) Pitting (**d**) Tooth profile (**e**) Tooth root fracture.

**Figure 9 sensors-22-04427-f009:**
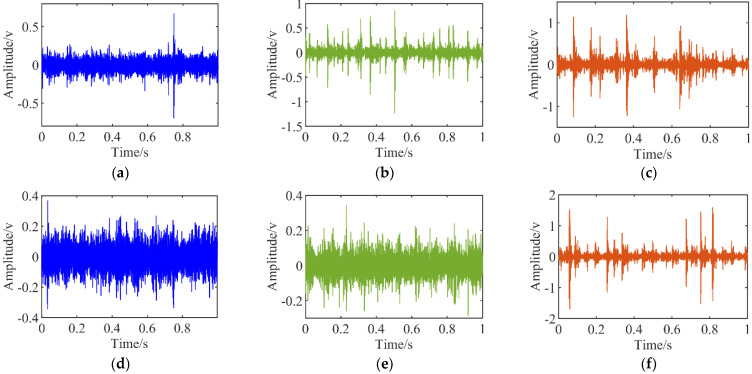
Signal time-domain waveform. (**a**) The normal; (**b**) Local wear; (**c**) Tooth profile; (**d**) Pitting; (**e**) Tooth root fracture; (**f**) Broken gear teeth.

**Figure 10 sensors-22-04427-f010:**
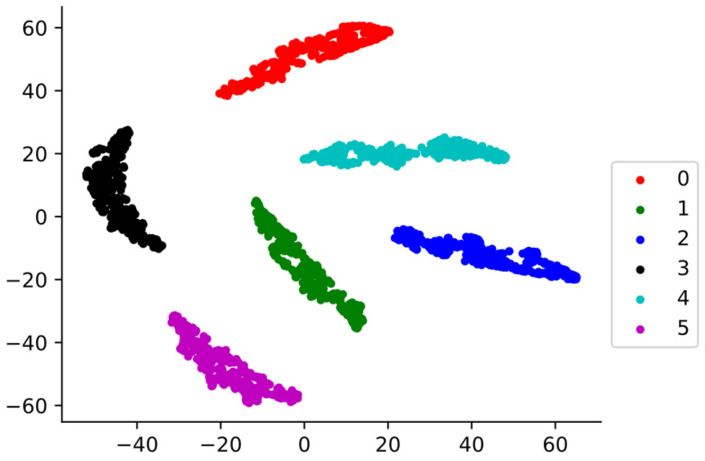
TSNE visualization results.

**Figure 11 sensors-22-04427-f011:**
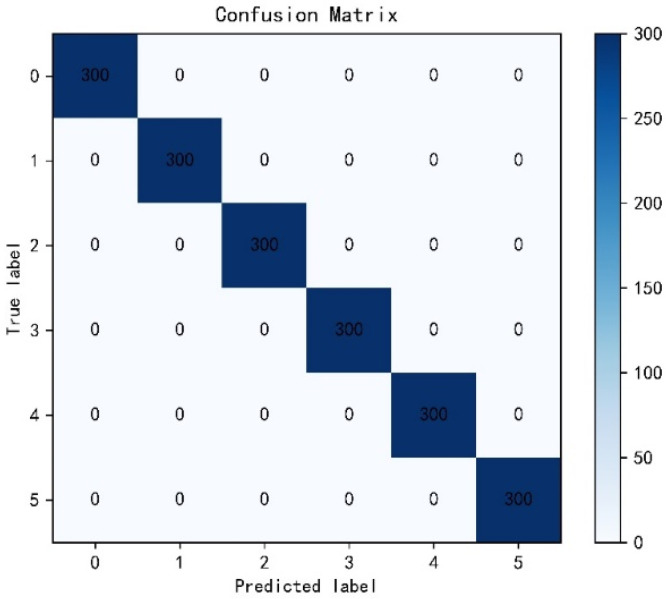
Experiment 1 confusion matrix results.

**Figure 12 sensors-22-04427-f012:**
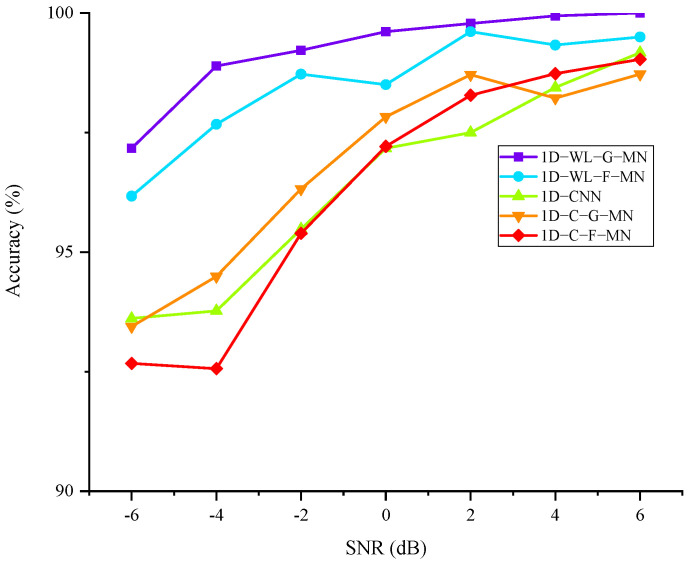
Comparison of anti-noise performance of the models.

**Figure 13 sensors-22-04427-f013:**
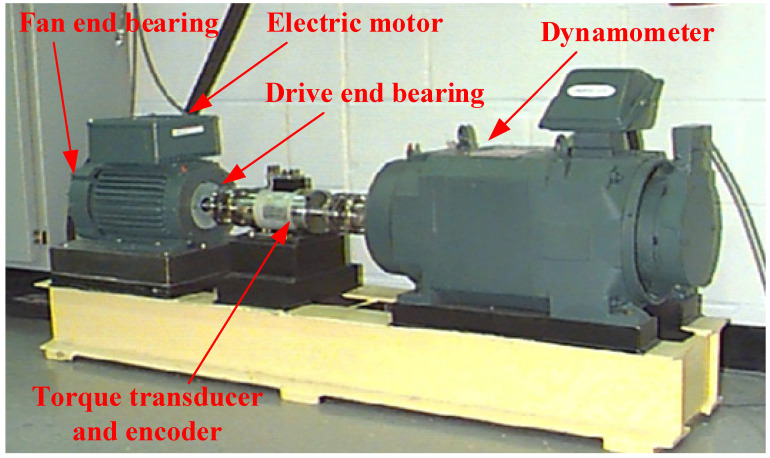
Bearing test bench of Western Reserve University.

**Figure 14 sensors-22-04427-f014:**
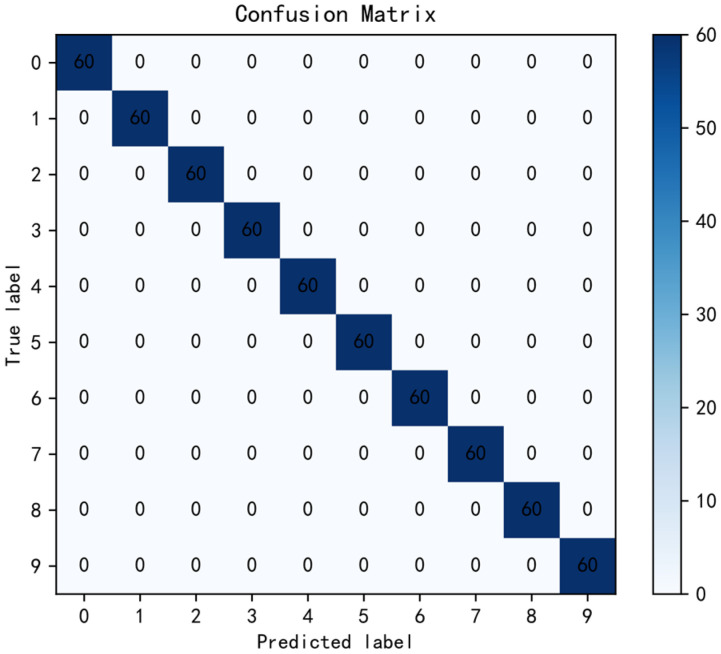
Experiment 2 confusion matrix results.

**Figure 15 sensors-22-04427-f015:**
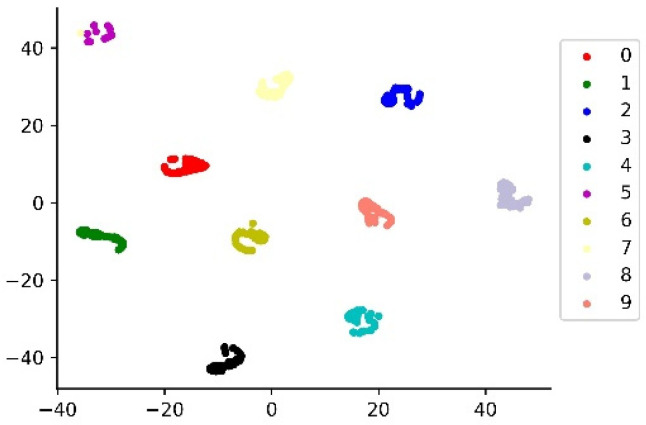
T-SNE visualization.

**Figure 16 sensors-22-04427-f016:**
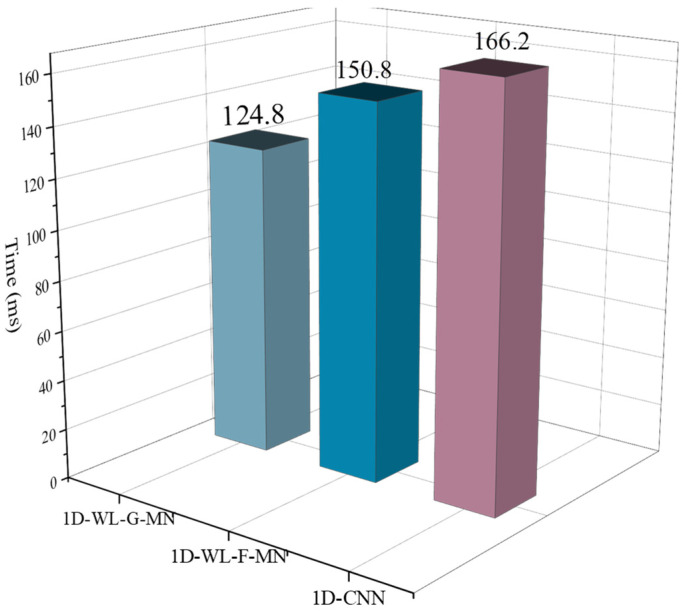
Running time in Raspberry Pi 4B.

**Table 1 sensors-22-04427-t001:** 1D-WL-G-MN network parameter.

Layer	Kernel Size (Length × Width)	Input Size (Length × Width)	Activation Function	Output Size (Length × Width)
Wavelet-Con1	55 × 1	2048 × 1	ReLu 6.0	2048 × 3
LPPool	2 × 1	2048 × 3	-	1024 × 3
DW1	5 × 1	1024 × 3	ReLu 6.0	1024 × 16
Maxpooling2	2 × 1	1024 × 16	-	512 × 16
DW2	5 × 1	512 × 16	ReLu 6.0	512 × 8
Maxpooling3	2 × 1	512 × 8	-	256 × 8
DW3	5 × 1	256 × 8	ReLu 6.0	256 × 6
GAP1	-	256 × 6	-	6 × 1
-	-	6 × 1	Softmax	6 × 1

**Table 2 sensors-22-04427-t002:** 1D-WL-G-MN network training parameter.

Parameter Name	Parameter Description
Training set:Test set	7:3
Optimizer	Adam
Batch size	70
Learning rate	0.001
Loss function	Cross-Entropy Loss

**Table 3 sensors-22-04427-t003:** Gearbox data set.

Fault Type	Number of Samples	Sample Length	Labels
The normal	1000	2048	0
Local wear	1000	2048	1
Tooth profile	1000	2048	2
Broken gear teeth	1000	2048	3
Tooth root fracture	1000	2048	4
Pitting	1000	2048	5

**Table 4 sensors-22-04427-t004:** Description of the different model structures.

NO.	Network	Description	Main Features
1	1D-CNN	One Dimensional-Convolutional Neural Network	1D Standard Convolutional Neural Network
2	1D-WL-F-MN	One Dimensional-Wavelet LPPool-FC-MobileNet	Network with wavelet convolution kernel and energy pooling in the first layer and a fully connected layer in the last layer
3	1D-C-F-MN	One Dimensional-CNN-FC-MobileNet	Network with standard convolutional kernel and maximum pooling in the first layer and a fully connected layer in the last layer
4	1D-C-G-MN	One Dimensional-CNN-GAP-MobileNet	Network with standard convolutional kernel and maximum pooling in the first layer and a global average pooling layer in the last layer
5	SVM	—	—

**Table 5 sensors-22-04427-t005:** Performance comparison of different models.

Model	Accuracy (%)	Parameter Quantity	Inference Time (s)
1D-WL-G-MN	99.94	1997	0.898
1D-WL-F-MN	99.28	11,219	1.380
1D-CNN	99.57	21,776	2.341
1D-C-G-MN	98.25	1197	0.910
1D-C-F-MN	99.13	11,219	1.351
SVM	41.27	-	-

**Table 6 sensors-22-04427-t006:** Bearing data set of Western Reserve University.

Test Subject	Load	Speed (r/min)	Number of Samples	Sample Length	Fault Type	Fault Diameter	Labels
6205-2RS JEM SKF	2HP	1750	200	2048	Normal	0	0
			200	2048	IF	0.007	1
			200	2048	IF	0.014	2
			200	2048	IF	0.021	3
			200	2048	OF	0.007	4
			200	2048	OF	0.014	5
			200	2048	OF	0.021	6
			200	2048	BF	0.007	7
			200	2048	BF	0.014	8
			200	2048	BF	0.021	9

**Table 7 sensors-22-04427-t007:** Model performance comparison.

Model	Accuracy (%)	Parameter Quantity	Inference Time (s)
1D-WL-G-MN	100.00	2037	0.886
1D-WL-F-MN	100.00	27,647	1.393
1D-CNN	99.61	39,932	2.459
1D-C-G-MN	99.52	2037	0.894
1D-C-F-MN	99.33	27,647	1.350
SVM	89.00	-	-

## Data Availability

The data presented in this study are available on request from the corresponding author.

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
