# Peer review of "An Improved MobileNet Network with Wavelet Energy and Global Average Pooling for Rotating Machinery Fault Diagnosis"

_sensors, 2022, doi:10.3390/s22124427_

Round 1

Reviewer 1 Report

The authors propose an improvement to the MobileNet network, based on the use of wavelet energy and GAP techniques, that allows them to reduce the number of needed parameters for the model, which reduce the computational effort, while maintaining the efficiency in results. The authors show how these improvements let apply the model to unidimensional signals in for fault diagnosis and detection, also facilitating the implementation of the model in embedded devices with low computational power.

The proposal is validated using experimental data of vibration signals, and the results are compared against other proposals of the MobileNet, a traditional full connected net, and Support Vector Machine, showing advantages such as a reduction in the number of parameters needed, reduction in processing time, while the precision is maintained, and sometimes improved. The authors also state that they have implemented the proposal into a Raspberry Pi device to validate the feasibility of implementing their proposal in embedded systems with low computing power, making it useful for implementation in real practical scenarios.

The results presented in the paper sound promising, however, it is necessary more clarity on how the authors present their work, and more evidence could be also useful to improve their work. In Section 3, a diagram with the structure of the proposed net is presented, and a brief description of the main elements of the net is provided; nevertheless, the explanation does not facilitate enough understanding for facilitating the reproduction of the work, which is an essential part of every scientific report. Therefore, I recommend authors to improve the description of the structure of the net, perhaps using more detailed diagrams that illustrate in a clearer way the interrelationship between the components of the net, the characteristics, and responsibilities of each component, as well as the ways those components interact with each other, for instance, using block diagrams or process or activities flow, or perhaps an algorithm.

In section four, the experimentation environment is described, and general data of the characteristics of the equipment used for experimentation are given, but no evidence is presented that allows observing the way in which the model was implemented in said equipment, which leaves open questions such as: Were only preprogrammed algorithms used in python? Was it necessary to make adaptations to integrate these algorithms? Should an algorithm have been implemented or modified to integrate the entire method?

At the end of section four, it is mentioned that the model was implemented in a Raspberry PI, and execution time data is presented, but it is not explained how these times were obtained, so that it can be verified how valid the data can be. To give greater credibility to the data presented, I recommend the authors to clearly explain how they captured the times they report, preferably showing evidence of the executions where those times are obtained.

Some minor details that should also be taken care of are:

Check the punctuation very carefully since semicolon (;) is frequently used instead of dot (.).

There are various mistakes and inconsistencies in the numbering of Figures and Tables, for example:

- the number of Figure 5 is repeated three times,

- there is not Figures 6 and 12

 - in line 230 Figure 4 is quoted, when it should be Figure 5.

- In some occasions the authors use Table and in others Tab. to number tables.

- Tab.3 is repeated in line 318

- In the headers of the tables, there are headers with the first letter in uppercase and others with lowercase

The phrase “Compared with the networks 1D-C-G-MN, 1D-C-F-MN and 1D-CNN that do not use wavelet energy processing, the proposed method has obvious improvement” is repeated.

Author Response

Dear Reviewer:

Thank you very much for giving us an opportunity to revise our manuscript. We greatly appreciate your constructive comments and suggestions on our manuscript sensors-1739176 entitled ‘An improved MoblieNet lightweight network fault diagnosis method based on wavelet energy and GAP’. Those comments are very helpful for revising and improving our paper, as well as the important guiding significance to other research. We have studied the comments carefully and made corrections which we hope meet with approval. The main corrections are in the manuscript and the responds to your comments are in a word document. The revised contents are marked in red in the revised manuscript.

Reviewer 2 Report

This paper is a great effort of the authors. However, I appreciate few more improvements:

  • The problem statement needed to be more precise in the abstract.
  • The contribution part in the Introduction shall be well identified and presented in specific manner.
  • What are the possibilities to incorporate explainable AI or explainablity of the feature spaces? I suggest including this article DOI: 10.3390/s21124070 in the literature and try to explore explainable AI block with the possibilities of deep learning with this.
  • Future directions should be well highlighted in the conclusion.
  • I want to see a few additional comparisons of following model architectures as well:
a.  Wang, B. and Li, H.S., 2021, September. Lane detection algorithm based on MoblieNet+ UNet lightweight network. In 2021 3rd International Symposium on Robotics & Intelligent Manufacturing Technology (ISRIMT) (pp. 352-356). IEEE. b.  Yao, Dechen, Guanyi Li, Hengchang Liu, and Jianwei Yang. "An intelligent method of roller bearing fault diagnosis and fault characteristic frequency visualization based on improved MobileNet V3." Measurement Science and Technology 32, no. 12 (2021): 124009.

Author Response

(The authors gave the same response as above.)

Reviewer 3 Report

Report on the manuscript "sensors-1739176" entitled "An improved MoblieNet lightweight network fault diagnosis method based on wavelet energy and GAP"

This manuscript proposes an investigation on network lightweight and its performance optimization based on MoblieNet networks. Experimental results are provided. Conclusions about this investigation are reported.

I have a good opinion about this work, which is relatively well written, its topic should be of interest to Sensors and its results are a suitable complement to the existing works. However, I think several aspects must be improved before it is suitable for publication, which I detail next:

1. The title of the manuscript needs to be rethought. I think it is too much to have seven words in "improved MoblieNet lightweight network fault diagnosis method".

2. The manuscript needs to be proofread by the authors. I have noted some drafting problems.

3. Words in the title are not usually in the keywords. In addition, the keywords are often written in alphabetical order. Please avoid using acronyms in the keywords.

4. The authors must check the use of all acronyms, abbreviations, and notations employed in the whole manuscript. Please define all the acronyms and abbreviations and then use them. Have in mind that the abstract is independent of the body of the paper, so that acronyms/abbreviations defined in the abstract (that are used there) must be again defined in the body of the paper. Please do not use capital letters in the text of an acronym/abbreviation. For example, "Complementary Ensemble Empirical Mode Decomposition (CEEMD)" must be "complementary ensemble empirical mode decomposition (CEEMD)".

5. A description of the sections of the manuscript must be added at the end of the introduction. For example: "The remainder of this paper is organized as follows. In Section 2, we... Section 3 describes... etc.".

6. Numbering of Subsection 3.1 is repeated. Please fix it.

7. The title of Subsubsections 4.2.1, 4.2.2, and 4.3.1 must be removed and the text inside them must be kept as run text in Subsections 4.2 and 4.3.

8. More details on how the data were obtained and pre-processed must be provided. A mention of the use of non-structured data must be informed. Today, close to 80% of the data generated by digital systems are non-structured.

9. The authors must provide more details about the computational framework employed in the manuscript. For example, software and packages utilized, features of the computer employed, runtimes, and other computational aspects must be added. 

10. The authors must summarize their methodology into an algorithm and/or in a flowchart so that the readers can follow it easier. Thus, the practitioners could have some guidelines when applying this methodology.

11. I do not have each numerical result in detail. I recommend the authors to check them.

12. In my opinion, the implications and results of the study are underdeveloped and must be improved and explained further in the final section.

13. Also, the authors must add limitations of the study and ideas for further research. Then, I suggest titling the final section: "Conclusions, limitations, and future research".

14. The authors must check whether all references are cited and all citations are in the reference list. 

Author Response

(The authors gave the same response as above.)

Round 2

Reviewer 1 Report

The paper has been significantly improved after the review, so I consider it is now acceptable for publication. However, there are some minor mistakes that should be corrected such as:
-    In Figure 4, authors are using the word date instead of data.
-    In line 359 in page 12, authors are citing Figure 5 instead of Figure 7.

Author Response

Dear Reviewer:

Thank you very much for giving us an opportunity to revise our manuscript. We greatly appreciate your constructive comments and suggestions on our manuscript sensors-1739176 entitled ‘An improved MoblieNet lightweight network fault diagnosis method based on wavelet energy and GAP’. Those comments are very helpful for revising and improving our paper, as well as the important guiding significance to other research. We have studied the comments carefully and made corrections which we hope meet with approval. The main corrections are in the manuscript and the responds to your comments are as follows . The revised contents are marked in red in the revised manuscript.

Reviewer 2 Report

Congratulations. Please update reference number 15. It is not in a correct format.

Author Response

(The authors gave the same response as above.)

Reviewer 3 Report

Accept in present form. The authors must remove the abbreviation "GAP" used in the title and write the full text related to GAP with no abbreviation.

Author Response

(The authors gave the same response as above.)
